# Effects of Beam Mode on Hole Properties in Laser Processing

**Tingzhong Zhang** [1,*], **Hui Li** [2], **Chengguang Zhang** [1] **and Aili Zhang** [3]

1    School of Mechanical and Electrical Engineering, Zhoukou Normal University, Zhoukou 466001, China; zhangtingzhong0814@126.com
2    Faculty of Engineering, Huanghe Science and Technology College, Zhengzhou 450006, China
3    Library, Zhoukou Normal University, Zhoukou 466001, China
*    Correspondence: ztz0416@zknu.edu.cn

**Abstract:** The laser beam mode affects the power density distribution on the irradiated target, directly influencing the product quality in laser processing, especially the hole quality in laser drilling. The Gaussian beam shape, Mexican-Hat beam shape, Double-Hump beam shape, and Top-Hat beam shape are four typical laser beam modes used as a laser heat source and introduced into our proficient laser-drilling model, which involves complex physical phenomena such as heat and mass transfer, solid/liquid/gas phase changes, and two-phase flow. Simulations were conducted on an aluminum target, and the accuracy was verified using experimental data. The results of the simulations for the fundamental understanding of this laser–material interaction are presented in this paper; in particular, the hole shape, including the depth–diameter ratio and the angle of the cone, as well as spatter phenomena and, thus, the formed recast layer, are compared and analyzed in detail in this paper. This study can provide a reference for the optimization of the laser-drilling process, especially the selection of laser beam mode.

**Keywords:** laser drilling; beam mode; melt pool; spatter; recast layer; hole geometry





## 1. Introduction

The laser, since its invention in 1960, has brought about revolutionary changes to our world due to its high energy density, high spatial and temporal coherence, and directional selectivity. Now, it is widely used in a variety of fields, such as scientific research and industrial manufacturing. However, thermal-based laser processing (drilling, welding, cutting, 3D printing, etc.) is complicated due to by-products such as spatter, recast, and cracking [1,2]. To restrain these disadvantages, the quality of the processed materials has become a hot topic. Important factors in laser material processing include the laser power, beam radius, pulse duration, defocusing distance, repetition rate, atmospheric pressure, and beam mode. Optimizing the machining parameters for the pulse duration, repetition rate, and combined pulse has been the focus of considerable interest [3–8]. A growing number of theoretical and experimental works have also investigated the effects of atmosphere pressure on the processed material [9–11].

However, the influence of beam mode on the laser–material interaction process is an area that receives less attention.

The laser beam mode is an important physical parameter that characterizes the power density distribution at the laser-irradiated target and has an important influence on the physical phenomena in the laser–material interaction. While most studies of laser–material interactions adopt a Gaussian beam shape (GS) for simplicity [12], one often finds several different shapes of laser beam, such as the Mexican-Hat beam shape (MH), Double-Hump beam shape (DH), and Top-hat beam shape (TH). Gerber and Graf [13] reported that TH is more suitable for some laser applications than GS. For example, in laser material processing, under certain conditions, the steep boundary of TH can produce sharp edges and flat bottoms in the processed workpiece. Campanelli et al. [14] studied the dimensions

of the molten tracks and the macro- and microstructure of the target during fiber laser surface re-melting with TH. Fabbro [15] studied Zn-coated steel sheets with CW Nd-Yag laser welding under TH and GS and obtained a keyhole geometry. They found that the rear keyhole wall was vertical for the TH beam shape as compared with that of GS; however, the efficiency using GS is much greater than those obtained with TH. Similarly, Kaplan [16] modeled fiber-guided laser deep-penetration welding and compared the keyhole shape and penetration depth under TH and GS. They found that the inlet surface of the keyhole obtained under TH was narrower and steeper compared to that of the GS. Kubiak et al. [17] studied the temperature field of targets during the laser heating process under GS and TH, respectively. The theoretically results demonstrated that the temperature fields were greatly influenced by laser beam intensity profiles. Lee and Mazumder [18] simulated the interaction between iron and a $CO_2$ laser with GS, MH, and DH, etc., and investigated the effects of laser beam mode on the melt pool. Wu and Zhang [19] conducted filler powder laser welding and studied the properties of heat and mass transfer during three types of laser–powder coupling. The results show that the temperature rising-up history and heating times and the heat-affected zone were influenced by the laser beam mode.

Currently, most numerical studies on the effects of the laser beam mode are focused on laser welding. However, the laser energy density is lower and the melt-flow characteristics are more gentle compared to those of laser drilling; these results show that this technique is not suitable for laser drilling. Moreover, currently, there is still a lack of knowledge about the effects of the beam modes on melt ejection, the recast layer and other flaw and hole properties during the laser-drilling process.

Doan [20] transformed GS to TH and DH via a novel laser beam shaper approach and conducted a laser-drilling experiment with the shaped laser beam. The experiment results revealed that the diameter of the drilled hole increased and the hole depth and heat-affected zone decreased in the order of GS, TH, and DH. Kim [21] conducted a SiC laser-drilling experiment in an air and water environment with DH and GS, respectively, and investigated the effect of the beam mode on the hole quality. Coutts [22] transformed GS to TH through a complex experimental system and conducted laser-beam-drilling experiments. The results revealed that the laser with TH could produce a through hole with a low wall taper, low eccentricity, and a minimal heat-affected zone, or a blind hole with a flat bottom. On the other hand, the GS has a higher peak power density than that of TH and is more suitable for laser deep-penetration welding and laser cutting. The aforementioned experiments required a custom-built experiment facility and were time consuming, and the principle of the thermal phenomena in laser drilling was still not clear. Therefore, theoretical study is necessary for a profound understanding of the effects of laser beam mode on a product's properties.

Han and Liou [23] constructed a laser–matter interaction model, based on which, they obtained the geometries and temperature fields of melt pools, as well as the flow patterns of melts, under different laser beam modes. Volpp [24] obtained the dynamic properties of the keyhole during aluminum laser beam welding with GS, DH, and TH based on a semi-analytical model. They found that the keyhole geometry and pore and spatter sizes were significantly influenced by the laser beam mode. However, analytical models are often based on some assumptions, which usually lead to the simulation results deviating from the actual results.

At the moment, there is no systematic study in the literature highlighting the effects of the GS, TH, DH, and MH on melt-flow phenomena, such as spatter, the recast layer, and the hole shape, during laser drilling. In this work, an experimentally verified numerical laser-drilling model [25,26] was tested on aluminum alloy, and the dynamic process of GS, TH, DH, and MH laser–matter interaction was studied.

## 2. Mathematical Model

When the laser beam irradiates the target surface, the material beneath the laser beam experiences a temperature rise, melting/solidification, vaporization, and splash phenomena. In order to more accurately reflect the law of physical action between the laser and material, a verified physical model including solid, liquid, and vapor phases

is used in this study to reveal the effects of the laser beam mode on the laser-drilling process, especially on the flow properties of the melt pool. The governing equations for the conservation of mass, momentum, and energy can be expressed in the following form. A finite element platform COMSOL was utilized to perform a discrete solution for Equations (1)–(4). The schematic diagram of laser drilling with boundary conditions is depicted in Figure 1. It comprises two phase flows and phase transitions. The level-set method (Equation (3)) is used to capture the free surface (air/target interface).

$$\nabla \cdot \boldsymbol{u} = m_0 \delta(\phi) \left( \frac{\rho_l - \rho}{\rho^2} \right) \tag{1}$$

$$\rho \left( \frac{\partial \boldsymbol{u}}{\partial t} + \boldsymbol{u} \cdot (\nabla \cdot \boldsymbol{u}) \right) = \nabla \cdot \left[ -pI + \mu \left( \nabla \boldsymbol{u} + (\nabla \boldsymbol{u})^T \right) \right] + \rho \boldsymbol{g} \beta_l (T - T_m) + K\boldsymbol{u} + \boldsymbol{F} \cdot \delta(\phi) \tag{2}$$

$$\frac{\partial \phi}{\partial t} + \boldsymbol{u} \cdot \nabla \phi - m_0 \delta(\phi)(\frac{\phi}{\rho_l} + \frac{1-\phi}{\rho_v}) = \gamma_{ls} \nabla \cdot (\varepsilon_{ls} \nabla \phi - \phi(1-\phi)\frac{\nabla \phi}{|\nabla \phi|}) \tag{3}$$

$$\rho C_p [\frac{\partial T}{\partial t} + \nabla \cdot (\boldsymbol{u}T)] = \nabla \cdot (\lambda \nabla T) + (Q_{\text{laser}} - m_0 H_v - \xi k_b (T^4 - T_0^4) - h(T - T_0)) \cdot \delta(\phi) \tag{4}$$

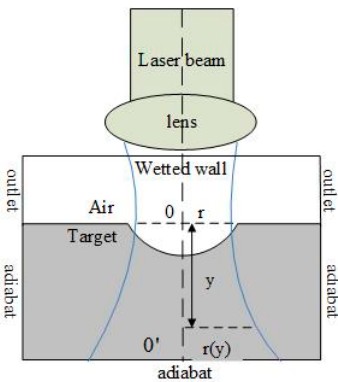

**Figure 1.** Schematic diagram of laser drilling and corresponding boundary conditions.

Here, in Equation (1), $\boldsymbol{u}$ is the fluid velocity [m/s], $m_0 = \sqrt{\frac{m}{2\pi k_b}} \frac{p_{sat}(T)}{\sqrt{T}} (1 - \beta_r)$ is the mass flux [kg/(m$^2$·s)], in which $m$ is the atomic weight of metal [kg], $k_b$ is the Boltzmann constant [J/K], and $\beta_r$ is the retro-diffusion coefficient assumed to be 0 at the beginning of evaporation, while 1 at steady state [25]. $\rho$ is the mixture density [kg/m$^3$], $\rho_l$ is the liquid density [kg/m$^3$], $\delta(\varphi)$ is the delta function, and $\varphi$ is the level-set function.

In Equation (2), $t$ is time [s], $p$ is pressure [Pa], $I$ is the unit matrix, T is the transpose of the matrix, $\mu$ is kinematic viscosity [N·s/m$^2$], $\boldsymbol{g}$ is gravitational acceleration [m/s$^2$], $\beta_l$ is the expansion coefficient [1/K], $T$ is temperature [K], $T_m$ is melting temperature of target material [K]. $K$ is the isotropic permeability in a porous medium model [2], describing the solid/liquid phase-change process. $\boldsymbol{F}$ is the gas/liquid interface force [N], including surface tension, recoil pressure, and Marangoni force [25], which mainly influences the properties of the fluid flow in the melt pool.

In Equation (3), here, capitalized and lower-case text is used for the level-set function, $\gamma_{ls}$ and $\varepsilon_{ls}$ are two level-set parameters, and they are all dimensionless numbers. $\nabla$ is the gradient function, and $\rho_v$ is the vapor density [kg/m$^3$]. The level-set equation is mainly used to capture the free surfaces of holes in the laser-processing process. Thereinto, the third term in Equation (3) is an additional added term that incorporates the mass loss due to evaporation into the model, so that the free surface of the hole can be more precisely captured.

In Equation (4), $C_p$ is the equivalent heat capacity [J/(kg·°C)], and $\lambda$ is the thermal conductivity [W/(m·K)]. $Q_{\text{laser}}$ is the laser irradiation thermal flux [W/m$^2$], and $m_0 H_v$ is the evaporation heat flow rate [W/m$^2$]. $\xi$ is the surface emissivity, and $h$ is the convection heat transfer coefficient. As can be seen, the second term on the right side of Equation (4)

also includes convection and radiation heat flux and heat dissipation at the free surface, which ensures the accuracy of temperature field calculation in the model.

Based on the mathematical Equations (1)–(4), boundary conditions should be discussed. For the pressure boundary, the outlet is $p = p_0$, the top side is a wetted wall, and the other sides is zero slip. For the thermal boundary, except for the air/target interface with intense heat exchange including laser energy input and heat dismission, the other sides of the metal target are adiabat, which can also be seen in Figure 1.

As a heat source, four modes of laser beam are adopted in this work. They are described as follows: Equations (5)–(8) are used for modeling GS, MH, DH, and TH, respectively, where $E$, $\tau$, and $x$ are the laser energy, the laser pulse width, and the distance to the laser beam axis, respectively.

$$\frac{2E}{\tau \pi r(y)^2} \exp\left(-\frac{2x^2}{r(y)^2}\right) \tag{5}$$

$$\frac{4E}{\tau \pi r(y)^2} \left(\frac{x^2}{r(y)^2}\right) \exp\left(-\frac{2x^2}{r(y)^2}\right) \tag{6}$$

$$\frac{2E}{\tau \pi r(y)^2} \left(1 - \frac{2x^2}{r(y)^2}\right)^2 \exp\left(-\frac{2x^2}{r(y)^2}\right) \tag{7}$$

$$\frac{E}{\tau \pi r(y)^2} rect(x) \tag{8}$$

The intensity profile curves are illustrated in Figure 2. In the simulation, the influence of beam divergence is also considered, the radius of the laser beam at a y distance away from the focal plane is calculated as r(y) = $r\,[1 + (y/y_f)^2]^{1/2}$, and the schematic diagram is depicted in Figure 1, where $y_f$ is the Rayleigh length, $r$ is the laser beam radius at the focal plane, $r$ = 0.0002 m, $E$ = 5 J, and $\tau$ = 1 ms.

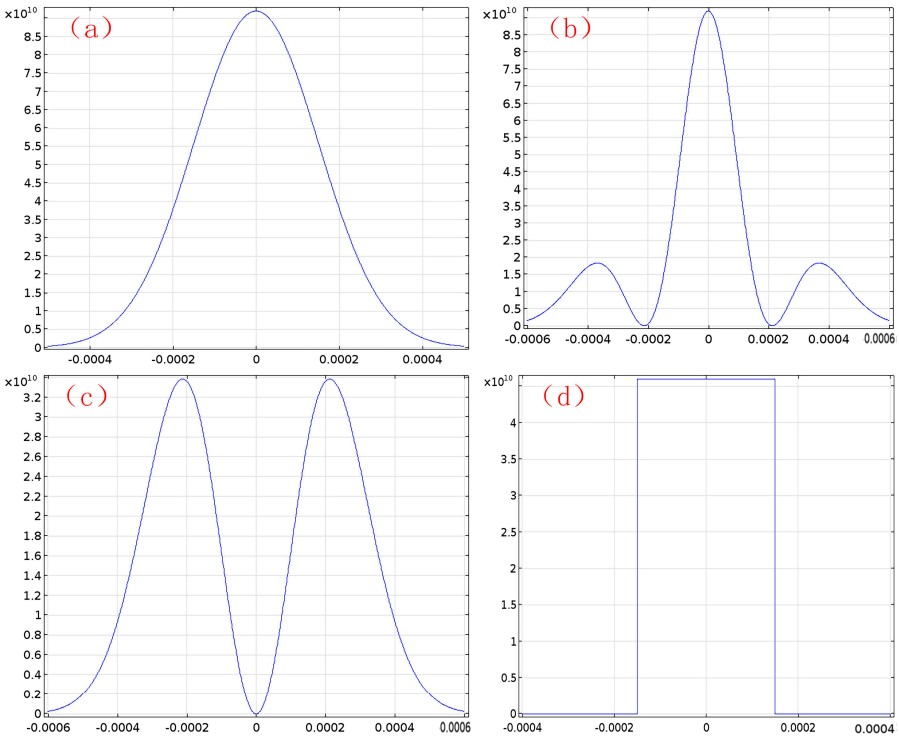

**Figure 2.** Intensity distribution of four different laser beam modes: (**a**) GS, (**b**) MH, (**c**) DH, and (**d**) TH along the *x* axis.

## 3. Results and Discussions

To further validate the established mathematical model, a comparison was made; refer to Figure 3a–c, between the predicted hole cross-section and the physical hole cross-section for three laser beam modes. The upside picture is a photomicrograph of the experimental hole cross-section, and the downside picture is the corresponding predicted hole cross-section. Very good agreement between the calculation and physical geometry was achieved, which indicates that the established model accurately captured the energy transfer occurring in the process. For the GS drilling case in Figure 3a, the predicted through-hole cross-section and the physical through-hole cross-section are presented, with a very reasonable agreement in dimensions. For the TH drilling case in Figure 3b, the predicted cross section and the physical cross section are presented as almost cylindrical, with a very reasonable agreement in dimensions, too. For the DH drilling case in Figure 3c, the predicted W-shaped cross section and the physical W-shaped cross section show almost no difference within the margin of error; this also indicates an accurate prediction of the thermal and flow field in the numerical model, which can then be used for the calculation of the spatter, recast layer, etc. Nevertheless, the prediction for the MH drilling case was not verified because of the complexity of the experiment system. However, the accuracy of the mathematical model can also be affirmed through comparisons of the other three modes. Below, the dynamic process of hole formation will be revealed in detail.

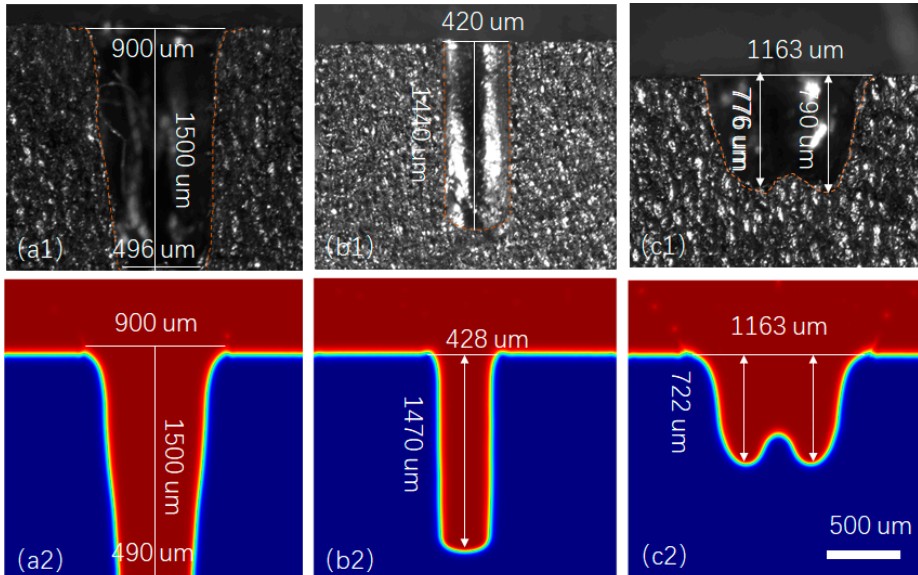

**Figure 3.** Comparison of the predicted and the physical hole cross sections for three different laser beam modes: (**a**) GS, (**b**) TH, and (**c**) DH.

Figure 4 shows the hole evolution process in the laser drilling of aluminum under GS mode. As can be seen in Figure 4a, a hole is formed and it is 0.265 mm in depth and 0.56 mm in diameter; in Figure 4b, the depth is 0.565 mm and the diameter is 0.624 mm; in Figure 4c, it is 0.874 mm in depth and 0.72 mm in diameter; in Figure 4d, it is 1.17 mm in depth and 0.78 mm in diameter; in Figure 4e, the target is almost penetrated and the hole has nearly reached 1.5 mm in depth and 0.852 mm in diameter; in Figure 4f, a penetrating hole is finished and the inlet diameter is 0.90 mm, the outlet diameter is 0.49 mm and the taper angle is about 15.9°, and the depth–diameter ratio is 2.16. From the above data, we can obtain a drilling velocity of about 1.67 m/s. In the hole formation process, we can see that the spatter image is faint due to chromatic aberration, and the distinct image can be viewed in the temperature field in Figure 5.

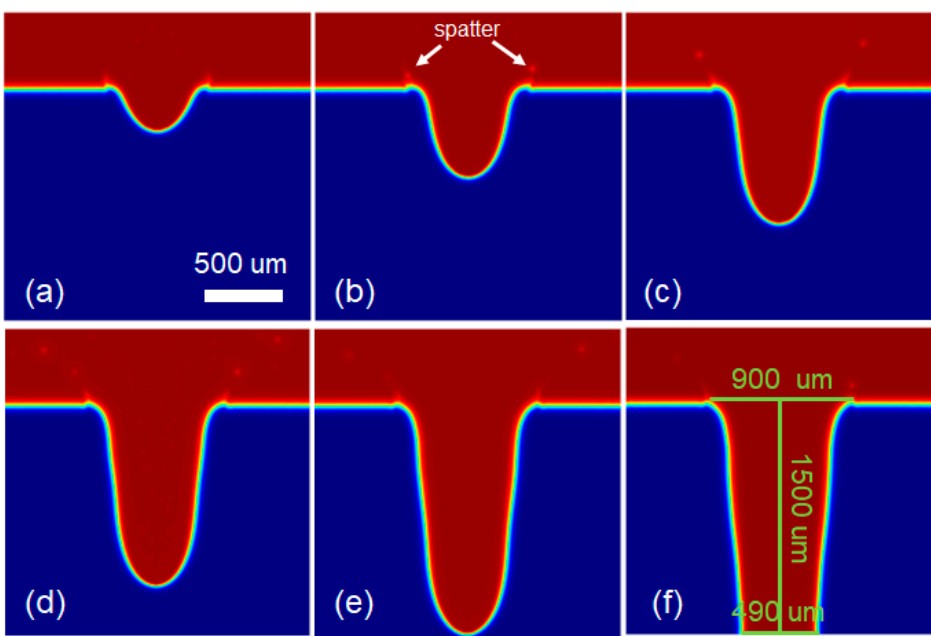

**Figure 4.** Sequences of penetrating hole at the moments of (**a**) 100 μs, (**b**) 300 μs, (**c**) 500 μs, (**d**) 700 μs, (**e**) 900 μs, and (**f**) 1000 μs in laser-drilled aluminum under GS mode. (The red is air, and the blue is the target).

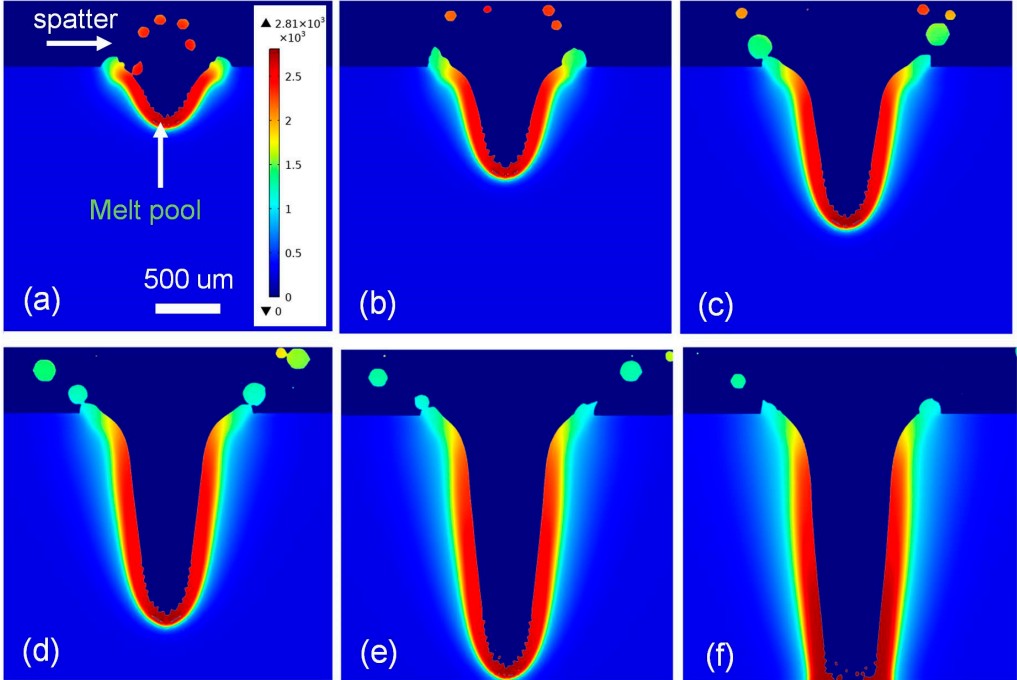

**Figure 5.** Sequences of the temperature field of the melt pool and spatter phenomena at the moments of (**a**) 100 μs, (**b**) 300 μs, (**c**) 500 μs, (**d**) 700 μs, (**e**) 900 μs, and (**f**) 1000 μs in the laser drilling of aluminum under GS mode.

Figure 5 shows the temperature field of the melt pool and spatter phenomena in the process of laser drilling aluminum under GS mode. As can be shown in Figure 5, the melt ejection almost exists in the whole hole evolution process. Additionally, the whole ejection process is abundant and can be mainly divided into three stages: in the first stage, shown in Figure 5a,b, the spatters are small and incandescence with a temperature of above 2500 K. Additionally, the ejection direction of the spatters is vertical to the target, mainly due to them coming from the middle of the inner liquid layer of the melt pool. In the second stage,

as shown in Figure 5c–e, a slightly larger spatter (the diameter is almost three times that of the initial spatter) is ejected from the melt pool, and the direction is nearly 45° from the target surface. However, the temperature of the bigger splash particle is a little lower, at about 1500 K. In the third stage, as shown in Figure 5f, the size of the spatter is smaller and the amount decreases, too. Furthermore, the temperature is about 1200 K, which is significantly lower than the melt pool temperature, 2500 K.

As can be concluded from the above results in Figure 5, the fiery spatter mainly comes from the high-temperature melt in the middle of the inner liquid layer along the hole wall; while the bigger and cooler spatter comes mainly from the top of the outer liquid layer along the hole wall. From a physical point of view, the former is mainly due to the recoil pressure from evaporation, which has overcome the surface tension, while the latter results from the upward flow of molten liquid along the hole wall under the Marangoni stress induced by the temperature gradient. Additionally, the flow field of the melt pool is described in detail in Figure 6, below.

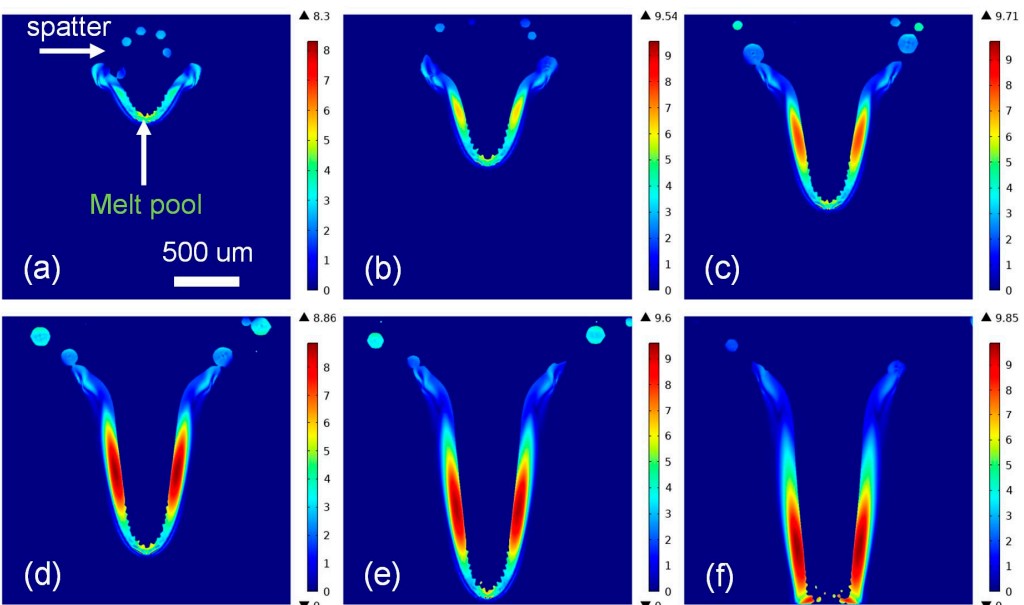

**Figure 6.** Sequences of the velocity field of the melt pool and spatter phenomena at the moments of (**a**) 100 μs, (**b**) 300 μs, (**c**) 500 μs, (**d**) 700 μs, (**e**) 900 μs, and (**f**) 1000 μs in the laser drilling of aluminum under GS mode.

Figure 6 shows the velocity field of the melt pool and spatter phenomena in the process of laser drilling aluminum under the GS mode. As can be seen in Figure 6a,b, in the first stage, the velocity of the outward and little spatter varies from 1.36 m/s to 2.55 m/s, and the maximum velocity of melt pool reaches 8.3 m/s with turbulence phenomena. As can be seen in Figure 6c–e in the second stage, the spatter velocity shifts from 1.38 m/s to 4.50 m/s, and the molten pool is relatively smooth. In the third stage, as Figure 6f shows, the spatter velocity decreases to 0.8 m/s due to the slow velocity of the upper liquid layer. However, generally speaking, the position of high-velocity molten liquid moves down with the hole formation process due to the downward heat source in the target as time goes by.

Figure 7 shows the recast layer formation process in the laser drilling of aluminum under GS mode. As shown in Figure 7a, accompanying the hole formation process in Figure 4a, the liquid layer clinging to the hole wall is formed and the overall feature is thick at the top and thin at the bottom, the bottom layer thickness is 0.03 mm, and the top maximum layer is 0.08 mm. As time goes by, as seen in Figure 7b–d, the bottom liquid layer remains unchanged in 0.03 mm, while the top thick liquid layer is 0.09 mm, 0.1 mm, 0.12 mm, and 0.13 mm, respectively, which shows a streamline changing rule. In general, in the laser-drilling process under GS mode, before the target is penetrated, the thickness

of the bottom liquid layer is almost unchanged at 0.03 mm, and the up-layer increases gradually from 0.09 mm to 0.13 mm. As can be seen in Figure 7f, a penetrating hole is finished, with the top liquid layer thickness being 0.14 mm, the middle area thickness being 0.132 mm (as indicated in Figure 7f), and the bottom liquid layer being 0.07 mm as laser drilling terminates. Due to heat convection, conduction, and radiation, the liquid layer along the hole wall gradually cools and eventually forms recast layer, which is consistent with the literature report in Ref. [27].

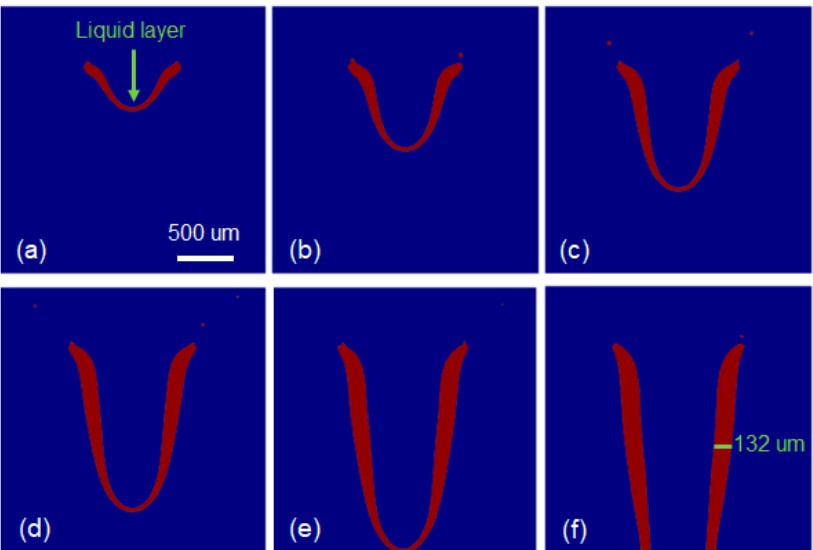

**Figure 7.** Sequences of the liquid layer (in red) with the hole evolution at the moments of (**a**) 100 μs, (**b**) 300 μs, (**c**) 500 μs, (**d**) 700 μs, (**e**) 900 μs, and (**f**) 1000 μs in the laser drilling of aluminum under GS mode.

Figure 8 shows the horn hole evolution process in the laser drilling of aluminum under MH mode. As shown in Figure 8a, the hole is formed and the hole depth is 0.25 mm and the inlet width is 0.312 mm. As can be seen in Figure 8b, the hole depth is 0.531 mm and the width is 0.446 mm; meanwhile, the marginal region of the hole becomes ridged slightly. As shown in Figure 8c, the hole depth reaches 0.812 mm and the entrance width of the hole is 0.464 mm; simultaneously, the wrinkle around the inlet of the hole begins to appear. With the passage of time, in Figure 8d, the upper neck is constricted and the inlet of the hole broadens; the largest width of the hole is 1.16 mm and the narrowest width of the hole is 0.312 mm. And then a horn hole is formed with the hole depth 1.1 mm. As shown in Figure 8e,f, the waist position downward and the horn hole are further developed. Finally, the horn hole is formed, about 1.5 mm in depth and 1.039 mm, 0.375 mm and 0.3 mm in diameter, respectively.

Figure 9 shows the temperature field of the melt pool and spatter phenomena in the process of laser drilling aluminum under MH mode. As shown in Figure 9a, a small amount of spatter, which occurred in the hole formation process in Figure 8a, is found (only two particles in red) above the molten pool and the temperature of the spatter particles reaches 2600 K. As shown in Figure 9a,b, there are bigger and cooler spatters ejected from the molten pool periphery. As can be seen in Figure 9d–f, with the horn-shaped (horn) hole formation process in Figure 8d–f, the spatter becomes violent and the main feature is that the ejecting particles from the middle of the molten pool are smaller and the temperature is higher, and the ejecting particles at the edge of the molten pool are larger and the temperature is lower. The temperature of the hot particles reaches more than 2500 K, while the temperature of the cold particles is about 1600 K. Additionally, the amount of spatter increases significantly in the process of Figure 9d–f, compared to that of Figure 9a–c.

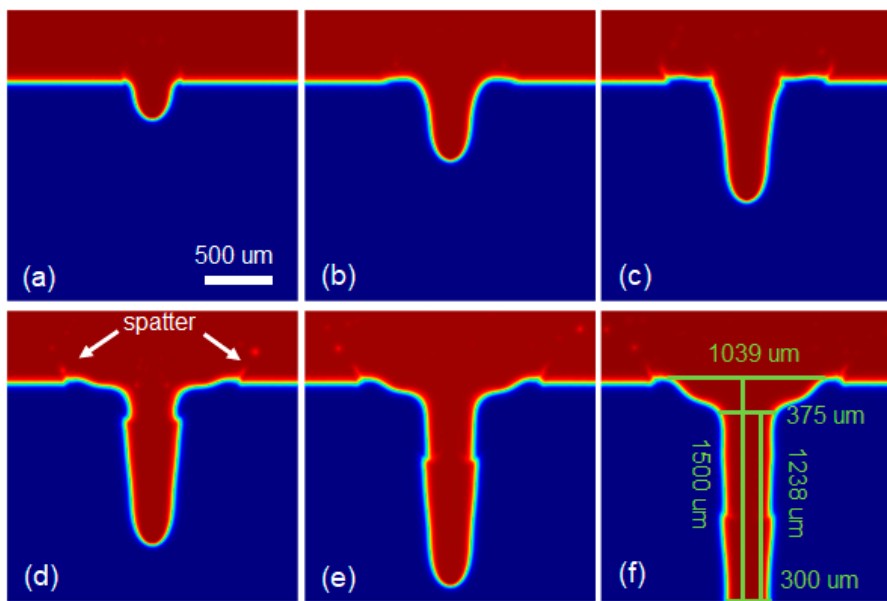

**Figure 8.** Sequences of the horn-shaped hole at the moments of (**a**) 100 μs, (**b**) 300 μs, (**c**) 500 μs, (**d**) 700 μs, (**e**) 900 μs, and (**f**) 1000 μs in the laser drilling of aluminum under MH mode. (The red is air and the blue is the target).

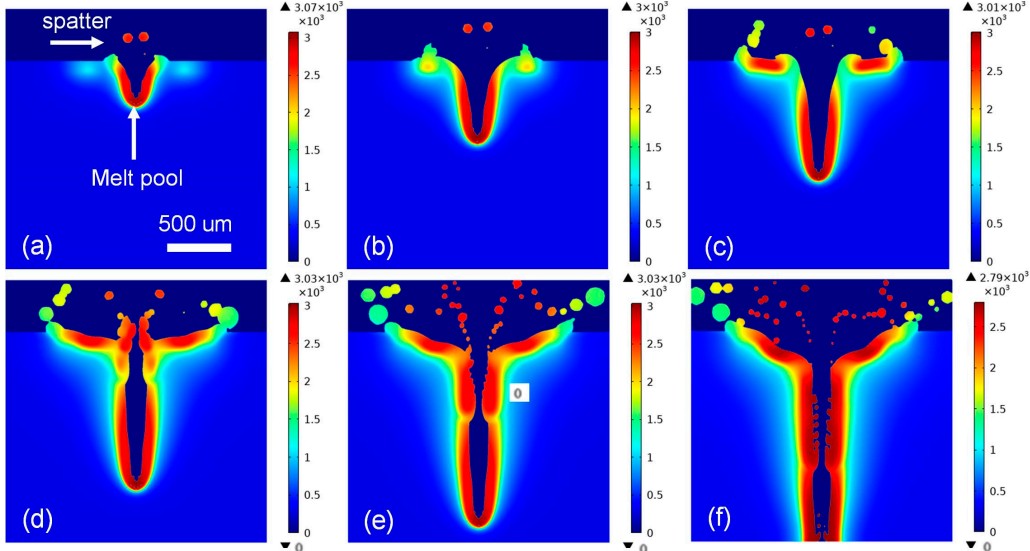

**Figure 9.** Sequences of the temperature field of the melt pool and spatter phenomena at the moments of (**a**) 100 μs, (**b**) 300 μs, (**c**) 500 μs, (**d**) 700 μs, (**e**) 900 μs, and (**f**) 1000 μs in the laser drilling of aluminum under MH mode.

Figure 10 shows the velocity field of the melt pool and the spatter phenomena in the process of the laser drilling of aluminum under MH mode. As shown in Figure 10a, accompanying the hole formation process in Figure 8a, the spatter is ejected from the melt pool at the velocity of 1.82 m/s; meanwhile, the maximum velocity of the liquid flow in the molten pool reaches 7.05 m/s. As can be seen in Figure 10b, the velocity of the surface fluid of the molten pool reaches 2.38 m/s and is about to break out of the molten pool and spill out. As shown in Figure 10c, the velocity of spatters varies from 1.37 m/s to 1.70 m/s and the maximum velocity of the fluid in the melt pool reaches 6 m/s. As time goes by, in Figure 10b–f, the maximum velocity of the fluid reaches about 12 m/s, and the velocity of the hot and small spatters reaches nearly 8.9 m/s, while the velocity of the cold and large spatters varies from 0.99 m/s to 2.89 m/s. From the above data, we can conclude that the large inertia force induced by the intense, high-velocity flow is the main cause of ejection.

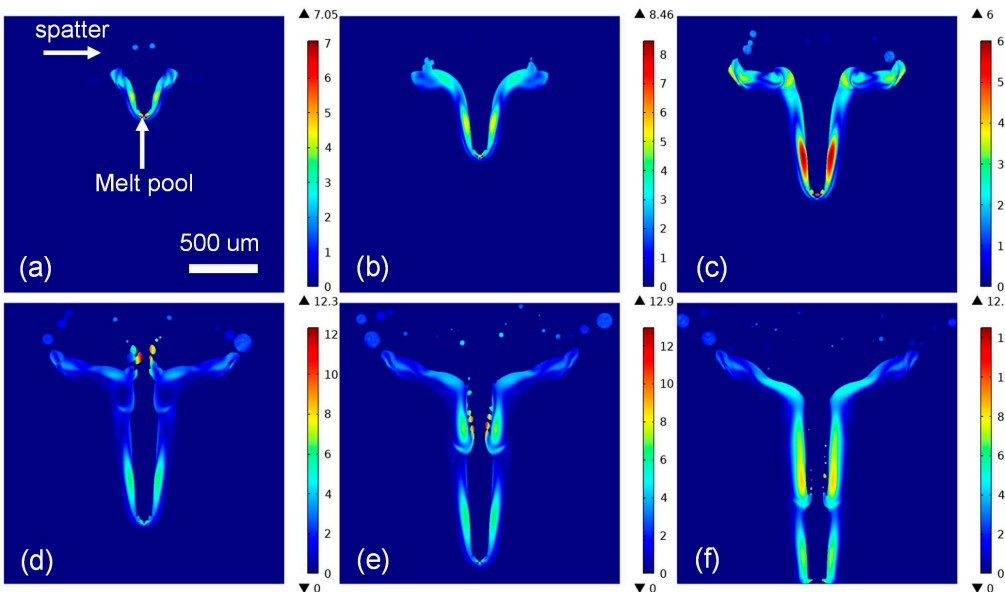

**Figure 10.** Sequences of the velocity field of the melt pool and the spatter phenomena at the moments of (**a**) 100 μs, (**b**) 300 μs, (**c**) 500 μs, (**d**) 700 μs, (**e**) 900 μs, and (**f**) 1000 μs in the laser drilling of aluminum under MH mode.

Figure 11 shows the horn-shaped recast layer formation process in the laser drilling of aluminum under MH mode. As shown in Figure 11a, accompanying the hole formation process in Figure 8a, the liquid layer moving along the hole wall and target surface are formed separately due to the unevenness of the MH laser intensity distribution. Due to the continuous laser irradiation, the two isolated melts join together, as shown in Figure 11b. And then, the thickness of the bottom and top of the liquid layer is about 0.028 mm and 0.11 mm, respectively. Accompanying the horn-shaped hole formation in Figure 8d–f, the horn-shaped liquid layer is formed, too. And, the overall key feature is that the liquid layer is thicker at the top and thinner at the bottom. Before going through the target, the thickness of the bottom layer is about 0.03 mm and the top maximum layer is 0.08 mm, 0.12 mm. When the hole is finished, as shown in Figure 11f, the thickness of the bottom and top of the horn-shaped liquid layer is about 0.075 mm and 0.147 mm, respectively. Due to thermal dissipation, such as heat convection, conduction, and radiation, the liquid layer adhering to the hole wall gradually cools and eventually forms the horn-shaped recast layer. It can be concluded that the thickness of the horn-shaped recast layer is thicker at the top and thinner at the bottom. Due to the heat convection, conduction, and radiation, the horn-shaped liquid layer adhering to the hole wall gradually cools and eventually forms a recast layer.

Figure 12 shows the hole evolution process in the laser drilling of aluminum under DH mode. As shown in Figure 12a, due to the laser pulse energy input, a double crater is formed initially, but the craters are separated from each other. Due to the continuous laser irradiation, as shown in Figure 12b, the double holes deepen, widen, and link together, forming an alphabetic-symbol-W-shaped hole of 0.11 mm in depth and 0.44 mm in width. As time goes by, as shown in in Figure 12c–f, the W-shaped hole further deepens and widens. The depth of the hole is 0.26 mm, 0.35 mm, 0.5 mm, and 0.65 mm, respectively. And the hole diameter is 0.51 mm, 0.58 mm, 0.61 mm, and 0.62 mm, respectively. As the laser–matter interaction terminates, the W-shaped hole is formed with a ratio of pit-depth to pit-diameter of about 0.69 and a cone angle of 42°.

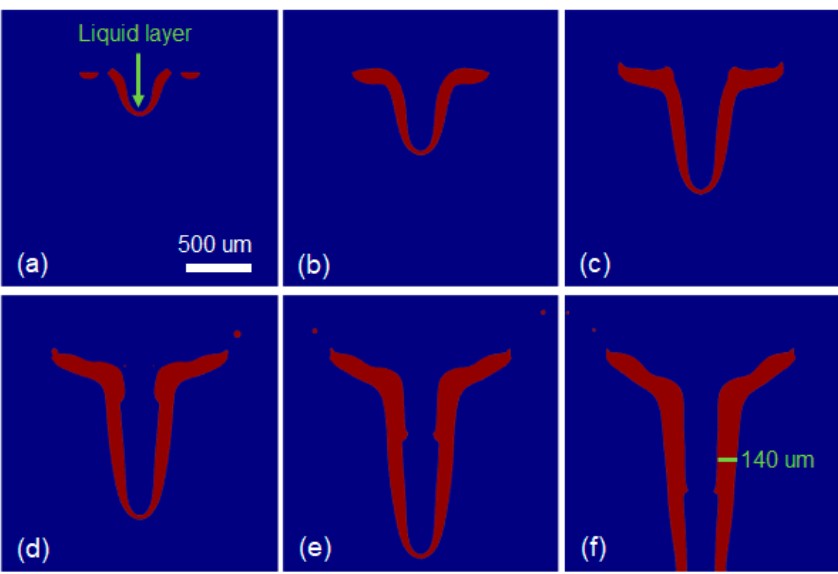

**Figure 11.** Sequences of the horn-shaped liquid layer (the red) with the hole evolution at the moments of (**a**) 100 μs, (**b**) 300 μs, (**c**) 500 μs, (**d**) 700 μs, (**e**) 900 μs, and (**f**) 1000 μs in the laser drilling of aluminum under MH mode.

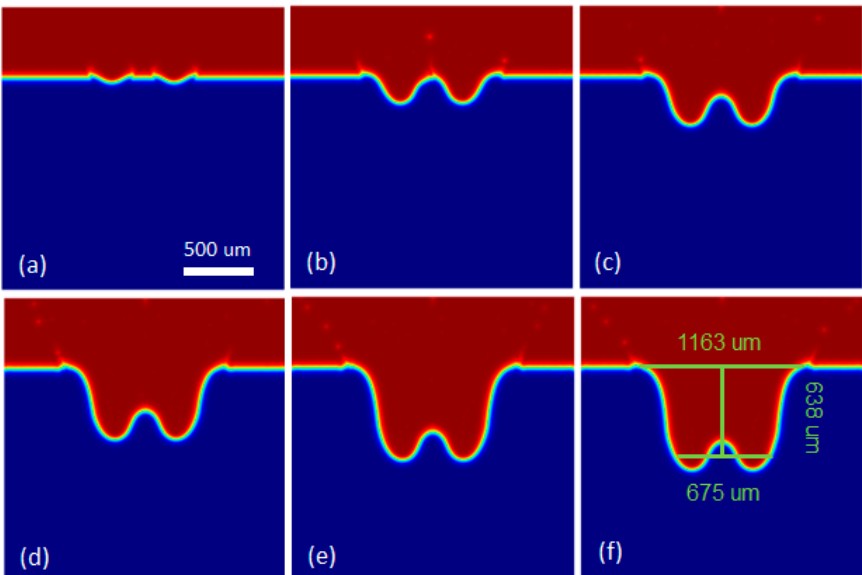

**Figure 12.** Sequences of the W-shaped hole at the moments of (**a**) 100 μs, (**b**) 300 μs, (**c**) 500 μs, (**d**) 700 μs, (**e**) 900 μs, and (**f**) 1000 μs in the laser drilling of aluminum under DH mode. (The red is air and the blue is the target).

Figure 13 shows the temperature field of the melt pool and spatter phenomena in the process of the laser drilling of aluminum under DH mode. As shown in Figure 11a, the irradiated material begins to melt and two identical high-temperature molten pools initially formed due to the camel-hump-shaped energy distribution in DH mode. With the continuous input of laser energy, the two identical high-temperature molten pools deepen, widen, and link together, forming a W-shaped molten pool. In addition, there are a large number of spatters ejected from the molten pool, which can be seen in Figure 13b. As time goes by in Figure 13c–f, accompanying the hole formation in Figure 12c–f, the W-shaped melt pool further deepens and widens, while splashing accompanies the entire pore-forming process. It should be made clear that the hot spatter particles with a temperature of about 2061 K generally come from the upper surface of the liquid layer at the bottom of the melt pool and are small in dimension. However, the cool spatter particles of about 1433

K from the edge of the molten pool are larger and the large spatter particles are about six times the diameter of the small particles.

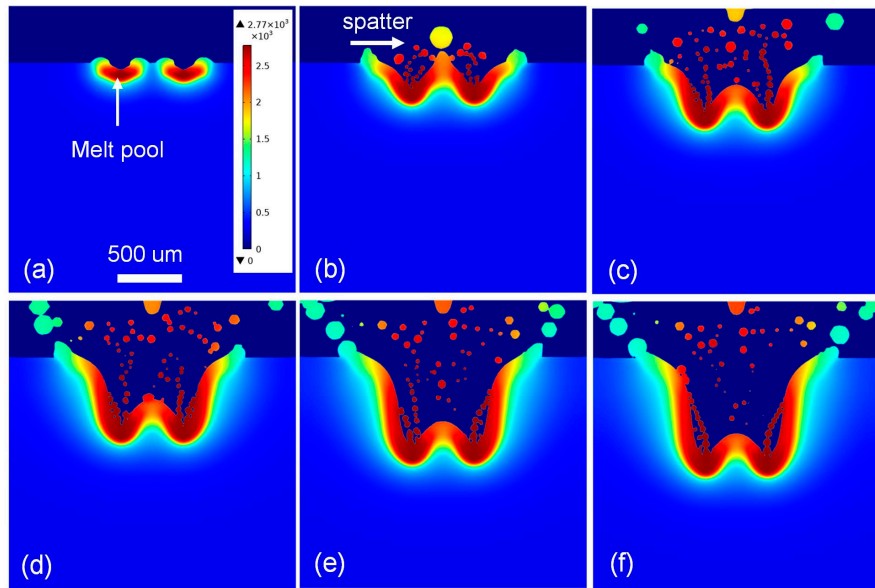

**Figure 13.** Sequences of the temperature field of the melt pool and star-shaped spatter phenomena at the moments of (**a**) 100 μs, (**b**) 300 μs, (**c**) 500 μs, (**d**) 700 μs, (**e**) 900 μs, and (**f**) 1000 μs in the laser drilling of aluminum under DH mode.

Figure 14 shows the velocity field of the melt pool and the star-shaped spatter phenomena in the process of the laser drilling of aluminum under DH mode. As can be seen in Figure 14a, the solid material under laser irradiation begins to melt and the liquid velocity varies from 2.54 m/s to 4.32 m/s. As the laser irradiation continues, as shown in Figure 14b, the molten liquid in the molten pool flows violently and the maximum velocity of the melt reaches about 15.1 m/s. Meanwhile, part of the melt material breaks away from the molten pool and forms spatter due to the large inertial force. The velocity of the spatter is from 1.22 m/s to 2.82 m/s. As time goes by, in Figure 14c–f, the maximum velocity of melt material is maintained at about 18 m/s and the star-shaped spatter phenomena is formed with velocities from 1.92 m/s to 9.53 m/s.

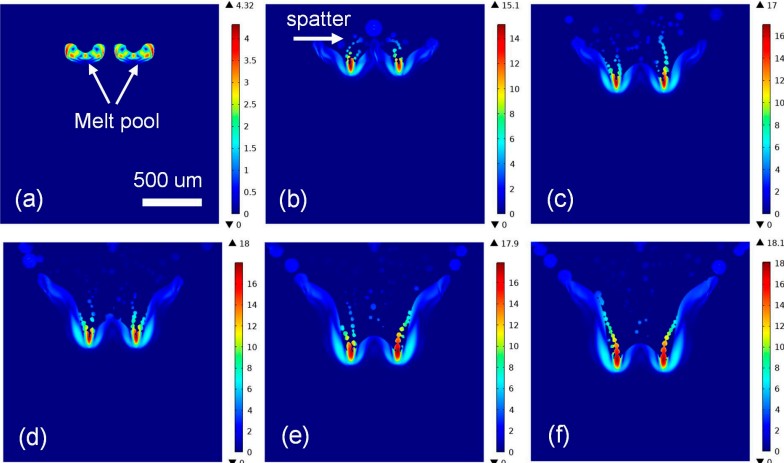

**Figure 14.** Sequences of the velocity field of the melt pool and the star-shaped spatter phenomena at the moments of (**a**) 100 μs, (**b**) 300 μs, (**c**) 500 μs, (**d**) 700 μs, (**e**) 900 μs, and (**f**) 1000 μs in the laser drilling of aluminum under DH mode.

Figure 15 shows the W-shaped recast layer formation process in the laser drilling of aluminum under DH mode. As shown in Figure 15a, accompanying the hole formation process in Figure 12a, two V-shaped liquid layers are initially formed; however, they are isolated from each other owing to the camel-hump-shaped energy distribution in DH mode. And then, the layer thickness is about 0.06 mm. As time goes by, as seen in Figure 15b, both of the V-shaped liquid layers join together, forming a W-shaped liquid layer. The maximum thickness of the liquid layer reaches 0.19 mm and the thickness at the narrowest point of the liquid layer is about 0.066 mm. In the W-shaped hole formation process in Figure 12c–f, the W-shaped recast layer is further developed and form the recast layer finally. As can be seen in Figure 15f, the thickness of the recast layer is uneven; the thickest part is about 0.185 mm and the thinnest part is 0.061 mm. Due to the heat convection, conduction and radiation, the W-shaped liquid layer adhering to the hole wall gradually cools and eventually forms a recast layer.

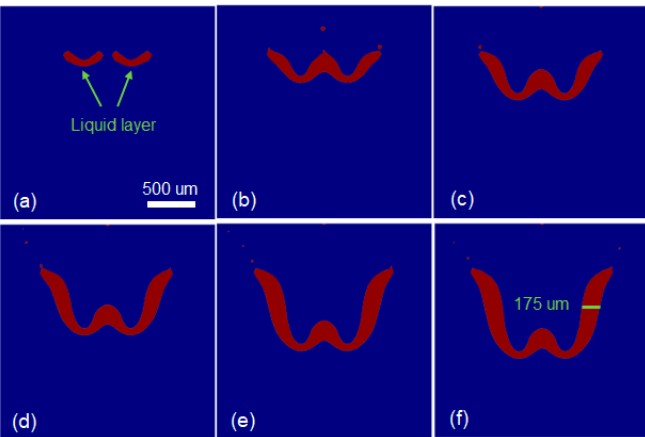

**Figure 15.** Sequences of the W-shaped liquid layer (red) with the hole evolution at the moments of (**a**) 100 μs, (**b**) 300 μs, (**c**) 500 μs, (**d**) 700 μs, (**e**) 900 μs, and (**f**) 1000 μs in the laser drilling of aluminum under DH mode.

Figure 16 shows the U-shaped hole evolution process in the laser drilling of aluminum under TH mode. As can be seen in Figure 16a, the hole is formed at 0.10 mm in depth and 0.36 mm in diameter. As time goes by, in Figure 16b–f, the depth is about 0.40 mm, 0.69 mm, 0.99 mm, 1.30 mm, 1.44 mm, respectively, and the width is 0.40 mm, 0.41 mm, 0.42 mm, 0.45 m, 0.45 mm, respectively. From the above data, we can obtain that the velocity of the U-shaped hole formation is about 1.45 m/s and the hole wall is nearly straight, with a cone angle of 4.7°. Furthermore, the depth–diameter ratio is 4.3.

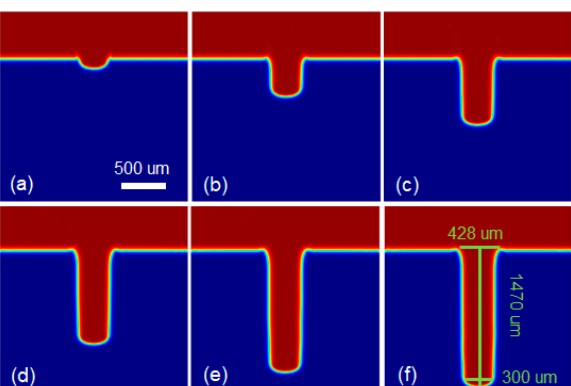

**Figure 16.** Sequences of the U-shaped hole at the moments of (**a**) 100 μs, (**b**) 300 μs, (**c**) 500 μs, (**d**) 700 μs, (**e**) 900 μs, and (**f**) 1000 μs in the laser drilling of aluminum under TH mode. (The red is air and the blue is the target).

Figure 17 shows the temperature field of the melt pool and the spatter phenomena in the process of the laser drilling of aluminum under TH mode. As can be seen in Figure 17, the melt ejection almost exists in the whole hole evolution process. However, the specific spatter state is different. As shown in Figure 17a, fiery spatter with a temperature of about 2622 K is ejected from the molten pool in a high temperature state. In Figure 17b, much smaller particles are ejected from the molten pool and a larger and cool spatter is about to splash. As time goes by, in Figure 17c–f, along with the evolution of the high-temperature molten pool, the vase-shaped spatter comes into view.

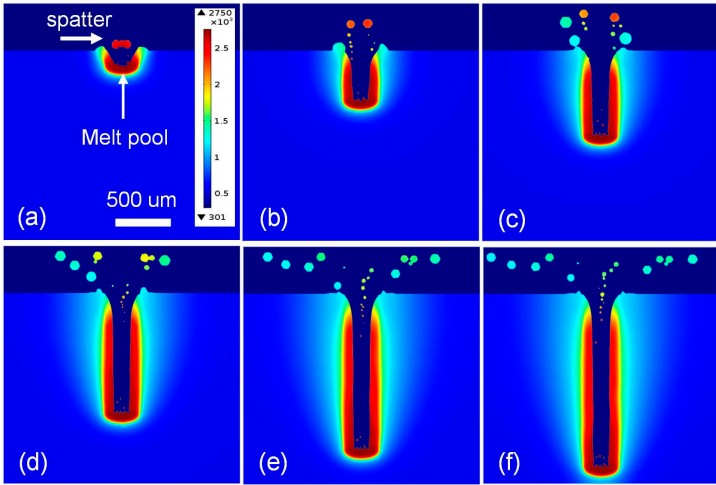

**Figure 17.** Sequences of the temperature field of the melt pool and vase-shaped spatter at the moments of (**a**) 100 μs, (**b**) 300 μs, (**c**) 500 μs, (**d**) 700 μs, (**e**) 900 μs, and (**f**) 1000 μs in the laser drilling of aluminum under TH mode.

Among the spatters, the highest temperature can reach 1959 K and the lowest temperature is about 1040 K. These spatters can be detrimental to the environment.

Figure 18 shows the velocity field of the melt pool and the vase-shaped spatter in the process of the laser drilling of aluminum under TH mode. As can be seen in Figure 18, in the hole formation process, the flow of the liquid layer is smooth and the velocity of the vase-shaped spatter is about 2.81 m/s.

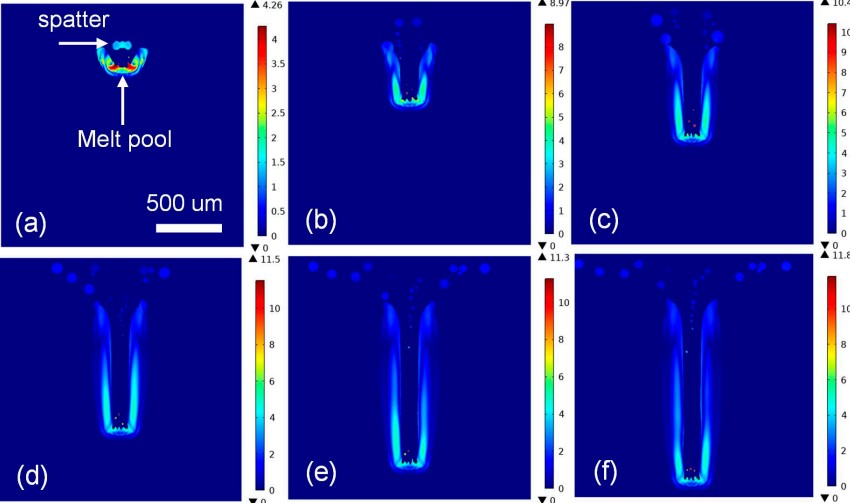

**Figure 18.** Sequences of the velocity field of the melt pool and the vase-shaped spatter at the moments of (**a**) 100 μs, (**b**) 300 μs, (**c**) 500 μs, (**d**) 700 μs, (**e**) 900 μs, and (**f**) 1000 μs in the laser drilling of aluminum under TH mode.

Figure 19 shows the U-shaped recast layer formation process in the laser drilling of aluminum under TH mode. As shown in Figure 19a, under the action of the recoil pressure induced by the metal vaporization due to the laser irradiation, the melt appears as a "U" shape moving along the hole wall. And then, the liquid layer is uneven with thickness about 0.046 mm. As time goes by, as seen in Figure 19b–f, the melt on both sides is streamlined, thicker in the middle and thinner on both sides; the thickest layer is about 0.085 mm, 0.109 mm, 0.126 mm, and 0.15 mm, respectively. The thinnest layer at the bottom remains about 0.55 mm thick. Due to heat convection, conduction, and radiation, the U-shaped liquid layer moving along the hole wall gradually cools and eventually forms a recast layer.

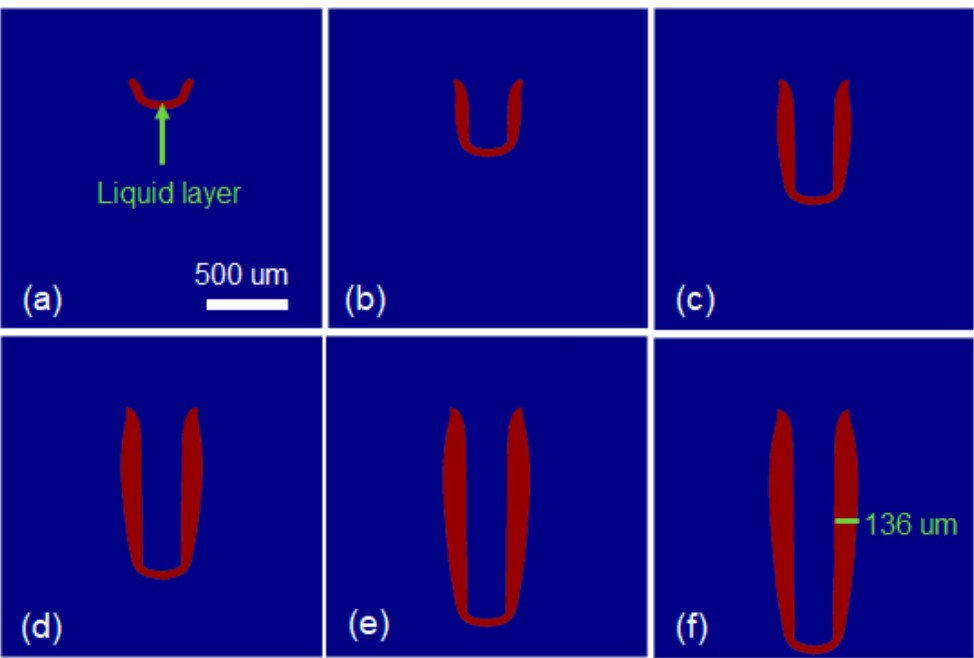

**Figure 19.** Sequences of U-shaped liquid layer (red) with the hole evolution at the moments of (**a**) 100 μs, (**b**) 300 μs, (**c**) 500 μs, (**d**) 700 μs, (**e**) 900 μs, and (**f**) 1000 μs in the laser drilling of aluminum under TH mode.

## 4. Conclusions

A sophisticated mathematical model is presented to investigate the effects of the laser beam mode on the laser–material interaction. Our simulation investigates the melt pool behavior, spatter phenomena, recast layer evolution, and, thus, hole formation in the laser drilling process, considering four laser beam modes: GS, MH, DH, and TH beam shapes. The velocity, the temperature, the liquid layer, and the drilled holes are presented. Four types of hole are observed for the corresponding laser beam mode. The relationship between the specific laser intensity distributions and the characteristic parameters of holes in the laser–material interaction process can be concluded as follows:

1. Firstly, the processing efficiency is a topic of great concern in modern industry. In terms of the drilling velocity for four laser beam modes, the order from high to low is GS, MH, TH, and DH, with velocities of 1.67 m/s, 1.53 m/s, 1.44 m/s, and 0.73 m/s, respectively.

2. Secondly, the quality of the hole is another topic that we care about. Here, we are mainly concerned with the cone angle of the hole and the depth–diameter ratio. In terms of the cone angle of the hole, the order from smallest to largest is 4.7° of TH, 15.9° of GS, 42° of DH, and 102° of MH, respectively. The last and most important one is the depth–diameter ratio, which is another very important indicator for measuring the quality of small holes. The order from largest to smallest for the depth–diameter ratio is 4.3 of TH, 2.16 of GS, 1.44 of MH, and 0.69 of DH, respectively.

3. Taking into account the environmental friendliness of laser processing, we mainly discuss the sputtering phenomenon here. The amount of spatter from low to high for the four laser beam modes is as follows: GS, TH, MH, and DH.

4. Finally, we still need to pay attention to the recast layer, because it directly affects the mechanical properties of products. With regard to the recast layer thickness, in order from thin to thick, it can be described as 132 μm of GS, 136 μm of TH, 140 μm of MH, and 175 μm of DH, respectively. (The above are average values, which are labeled in the images).

From our Discussion, the following conclusions can be obtained. As compared to the incomplete hole shape in the MH and DH laser-drilling process and the large cone angle of the hole in GS laser drilling, the TH laser-drilled hole has a high depth–diameter ratio, a small taper angle with little spatter ejection, and a relatively thin recast layer. The author hopes that the study can provide assistance in the laser-processing field, especially in beam mode selection.

**Author Contributions:** Methodology, H.L.; Investigation, C.Z.; Writing—original draft, T.Z.; Writing—review & editing, A.Z. All authors have read and agreed to the published version of the manuscript.

**Funding:** This research was funded by the Science and Technology Research Project of Henan Province, China [Grant Numbers: 222102220022, 242102311240].

**Institutional Review Board Statement:** Not applicable.

**Informed Consent Statement:** Not applicable.

**Data Availability Statement:** Data are contained within the article.

**Conflicts of Interest:** The authors declare no conflict of interest.

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
