# Peer review of "Effects of Beam Mode on Hole Properties in Laser Processing"

_coatings, doi:10.3390/coatings14050594_

Round 1

Reviewer 1 Report

Comments and Suggestions for Authors

The authors present simulations of laser drilling in the melt expulsion regime.
A set of four different spatial distributions for the laser intensity are investigated by showing simulation results for selected parameter settings.
In particular, the velocity of the drill base, the conical shape of the drilled walls as well as spatter and recast are properties considered.

It's a pity that the authors did not mention the boundary conditions for their model (lack of mathematical rigor) and did not mention the software used for their simulation (lack of numerical rigor).

Although, the authors address relevant properties of the drilling process the common know how about the topic is far more advanced (see reference 1,2) below) reporting about a software which is able to reproduce real drillings up to few percent with respect to the properties considered.

The authors considered an oversimplified setup neglecting the divergence of the radiation, neglecting the measured intensity distribution of real laser radiation as well as neglecting the feedback of ejected molten material onto the radiation propagation.

Unfortunately, the authors did not know about the fact, that the transient drill shape approaches an asymptotic shape (see reference 1,2). For pulse durations at which the asymptotic form is not achieved, the results show a large deviation in the results of successive holes, as the ejected material flows in the direction of the beam path and leads to scattering and absorption. For example, reference 1,2 of this review reveals that a cylindrical shape of the drilled wall can be achieved with real laser radiation by adjusting focal position and divergence angle, which both are not covered by the authors model.

typewriting:
line 24: word -> world
line 24: good direction -> directional selectivity
line 33: works -> work
line 35: characteristics -> characterises
line 106: Capitalised and lower case is used for the level set function
line 401: flows -> follows
line 402: hoe -> hole

faulty judgements:

line 35: "However, the influence of beam profile on laser-material interaction process is an area that receives less quantitative attention." In contrast to the authors judgement the literature cited [14-24] is already quantitative.
Additionally, see some theoretical work also quantitative, for example:
1)Interaktive Prozesssimulation für das industrielle Umfeld am Beispiel des Bohrens mit Laserstrahlung, ISBN: 978-3-86359-614-9
2)Reduced Modelling for Laser Drilling Process in Melt Expulsion Regime, ISBN.de, ISBN-10 3-98555-013-1

line 116: "Here, it is assumed that the beam shape does not change along the propagation direction"
See references 1,2 which demonstrate the influence of beam divergence and focal position on the shape of a borehole.

line 180: "Figure 5 shows the recast layer formation process in laser drilling".
Already in 2021 a PhD thesis 1,2 carried out a detailed analysis and simulation about these topics and should cited at least.

line 403: "The TH laser drilled hole has high depth-diameter ratio, small taper angle with little spatter ejection and relatively thinner recast layer. Which will be the recommended laser mode for laser drilling in future."
It's a pity that only simple mathematically defined and non-divergent beam shapes are considered in spite of measured, real distributions of laser radiation. It remains unclear, why a tophat distribution produces strong enough driving forces for the melt to be ejected, since only a gradient of the pressure from evaporation can accelerate the molten material at the drill base.

Comments on the Quality of English Language

Minor revision of typing errors are recommended

Author Response

Dear prof. XX and dear reviewers

Re: Manuscript ID: coatings-2984385 and Title: Effects of beam mode on hole properties in laser processing

Thank you for your letter and the reviewers’ comments concerning our manuscript entitled “coatings-2984385”. Those comments are valuable and very helpful. We have read through comments carefully and have made corrections. Based on the instructions provided in your letter, we uploaded the file of the revised manuscript. Revisions in the text are shown using red highlight for additions, and strikethrough font for deletions. The responses to the reviewer's comments are marked in highlight colors and presented following.

We would love to thank you for allowing us to resubmit a revised copy of the manuscript and we highly appreciate your time and consideration.

Sincerely.

Tingzhong zhang.

Reviewer 2 Report

Comments and Suggestions for Authors

Review of the manuscript Effects of Beam Mode on Hole Properties in Laser Processing by Zhang Tingzhong, Li Hui, Zhang Chengguang, Zhang Aili.

The paper presents simulation results of the laser drilling process to fundamentally understand this laser-material interaction, especially the shape of the hole, including depth-to-diameter ratio, cone angle, etc.

The work clearly has important practical and theoretical significance. But there are some notes:

1. Page 3 provides a mathematical model used to simulate processes. The above equation is used with reference to the literature [25,26]. To check the adequacy of the equation, it is necessary to indicate the units of measurement of the quantities included in the equation.

2. It is not specified what software was used for the simulation.

3. The work presents only the results of mathematical modeling. Since there are no comparisons with experimental results, it is not clear how adequate the mathematical model used is.

I think the work needs improvement.

Author Response

(The authors gave the same response as above.)

Round 2

Reviewer 1 Report

Comments and Suggestions for Authors

The quality of the revised version is acceptable and I recommend publicatio in the present form.

Reviewer 2 Report

Comments and Suggestions for Authors

Thanks to the authors for the changes made to the text of the manuscript. In this form the work can be published.